# Gradients protection in federated learning for Biometric authentication

## Abstract

In the context of face recognition models, different facial features contribute unevenly to a model's ability to correctly identify individuals, making some features more critical and, therefore, more susceptible to attacks. Deep Gradient Leakage (DGL) is a highly effective attack that recovers private training images from gradient vectors, posing significant privacy challenges in distributed learning systems where clients share gradients. Data augmentation, a technique for artificially manipulating the training set by creating modified copies of existing data, plays a crucial role in improving the accuracy of deep learning models.

In this paper, we explore various data augmentation methods to protect original training images, in test time thereby enhancing security in distributed learning systems as well as increasing accuracy during training. Our experiments demonstrate that augmentation methods improve model performance during training on augmented images, and we can use the same methods during testing as perturbation methods to preserve some features of the image and have safety against DGL.

This project has four primary objectives: first, to develop a vision transformer face validation model that trains on distributed devices to ensure privacy; second, to utilize augmentation methods to perturb private images and increase neural network safety; and third, to provide protection against attacks, ensuring that reconstructing attacks cannot extract sensitive information from gradients at any point in the system. and lastly we introduce a new novel perturbation method for a multi biometric authentication, system which offers accuracy for identification and guarantees safety and anonymity of entities.

## 1 Introduction

Federated learning (Rodríguez-Barroso et al., 2023) proposes letting participants train on their own data in a distributed fashion and share only model updates —i.e., gradients with the central server. The server aggregates these updates (typically by averaging) to improve a global model and then sends updates to participants. This process runs iterative until the global model converges. Merging information from individual data points into aggregated gradients intuitively preserves privacy to some degree.

While this setting might seem safe at first glance, a few recent works have begun to question the central premise of federated learning - is it possible for gradients to leak private information of the training data? Effectively serving as a "proxy" of the training data, the link between gradients to the data in fact offers potential for retrieving information: from revealing the positional distribution of original data (Melis et al., 2019; Shokri et al., 2017), to even enabling pixel-level detailed image reconstruction from gradients (Geiping et al., 2020; Zhao et al., 2020; Zhu et al., 2019).

Building on the findings in Wei et al. (2022), we draw two key insights: (1) different features contribute unequally to the task, with some being so insignificant that they can be masked without significantly affecting the model's performance. In their work, classification tasks were completed with high accuracy even with certain masked features, a concept we apply to our authentication task; and (2) There is an inherent trade-off between the model's security, robustness, and accuracy; it is essential to carefully balance these aspects in designing effective defenses.

Generalizability refers to a model's ability to maintain consistent performance when applied to new,

unseen data (testing data), as compared to its performance on the data it was trained on. Models with poor generalizability tend to overfit the training data. performing well on examples but struggling with new ones.(Shorten & Khoshgoftaar, 2019)

Data Augmentation is a very powerful method of achieving generalizability. The augmented data will represent a more comprehensive set of possible data points, thus minimizing the distance between the training and validation set, as well as any future testing sets. Through augmentation methods; more information can be extracted from the original dataset(Krizhevsky et al., 2012). By employing augmentation techniques, we can shift the model's focus from one part of an image to another. Furthermore, data augmentation can serve as a tool to increase the security of raw data during the training and testing phases. In this work, we examine various augmentation methods and their impact on model performance. Additionally, we explore how these methods can be leveraged to safeguard private information within a federated learning setup during test time.

Vision Transformer (ViT) Alexey (2020) is an architecture inherited from Natural Language Processing (Vaswani, 2017) while applied to image classification with taking raw image patches as inputs. Different from classical Convolutional Neural Networks (CNNs), the architectures of ViTs are based on self-attention modules (Vaswani, 2017), which aim at modeling global interactions of all pixels in feature maps. More precisely, ViTs take sequential image patches as inputs, and the attention mechanism enables interaction and aggregation directly among patch information. Therefore, compared to CNNs where image features are progressively learnt from local to global context via reducing spatial resolution, ViTs enjoy obtaining global information from the very beginning. Up till now, such convolution-free networks have been achieving great success on various computer vision tasks, including image classification (Touvron et al., 2021; Wu et al., 2021; Chen et al., 2021a; Li et al., 2022; Mao et al., 2022; Yao et al., 2023), object detection (Liu et al., 2021; Li et al., 2022; Yao et al., 2023), semantic segmentation (Strudel et al., 2021; Liu et al., 2021; Yao et al., 2023) and image generation (Chen et al., 2021b).

In this work, we utilize Vision Transformer (ViT) models Chen et al. (2023) instead of traditional Resnet face recognition models to achieve superior image mapping during training. This approach aims to enhance performance on an unseen dataset that significantly differs from the training set. Distributed training and collaborative learning have been widely used in large scale machine learning tasks. In most scenarios, people assume that gradients are safe to share and will not expose the training data. Some recent studies show that gradients reveal some properties of the training data, for example, property classifier Melis et al. (2019) (whether a sample with certain property is in the batch) and using generative adversarial networks to generate pictures that look similar to the training images (Gentry, 2009; Hitaj et al., 2017; Melis et al., 2019).

In the work Zhu et al. (2019), present an optimization algorithm that can obtain both the training inputs and the labels in just few iterations. The method deep leakage is an optimization process and does not depend on any generative models; therefore, DLG Zhu et al. (2019) does not require any other extra prior about the training set, instead, it can infer the label from shared gradients; the results produced by DLG (both images and texts) are the exact original training samples instead of synthetic look-alike alternatives(Yin et al., 2021). We propose a new VIT model with multiple attention windows to perform better on unseen data and seeks to focus and learn different features better. Our proposed augmentation method jointly seek to find (1) the optimal mask for deciding how much of the inputs to reveal versus conceal in the given region and (2) a trade off between security and accuracy of face recognition. Based on the idea of mixing images to learn the individual better (Kim et al., 2020) we propose to mix two biometrics to have better accuracy in authentication by having dynamic weight for each and try to find the optimal mask of the two inputs.

## 2 RELATED WORK

There is a rich literature for defense strategies against gradient attacks in distributed models. Most techniques focus on protecting the image as a whole ratter than focusing on the features. Each of these techniques has its own flaws. In this work we study, how separating features helps preserve accuracy in face authentication task, and also protect the original image. In order to guarantee privacy, it is necessary to introduce randomness to the learning algorithm.

Gradient perturbation Zhu et al. (2019); Sun et al. (2020), directly prunes the shared gradients in federated learning to defeat gradient leakage attacks; However, recent contributions Huang et al. (2021) found that it is usually required to prune too much gradient information to fully defeat gradient leakage attacks, which will greatly hurt the model accuracy.
Input data encryption Huang et al. (2020a;b) encrypts the data and hides private information; However, current state-of-the-art encryption methods Huang et al. (2020b) can also be evaded by adaptive attack methods (Carlini et al., 2021). Therefore, an effective defense method which can reliably protect the privacy of clients while preserving model accuracy is still highly demanded.

Zero-knowledge proofs (ZKPs) (Bonawitz et al., 2016; McMahan et al., 2017) are widely recognized cryptography tools that enable secure and private computations while safeguarding the underlying data. In essence, ZKPs empower a proof to convince a verifier of a specific fact without revealing any information beyond that fact itself. By verifying these proofs, the users can ensure the aggregator's actions are transparent and verifiable, installing confidence that the aggregation process is conducted with utmost honesty. ZKP (Geiping et al., 2020) can ensure the server is protected from aggregating malicious updates and constantly prove the transparency of users and server's action but unfortunately it is necessary to share the private raw information with server and trust the infrastructure and system as entity. Additionally, generating the proofs can be computationally expensive, impacting performance due to the time required for processing.

Input mixup creates virtual training examples by linearly interpolating two input data and the corresponding one- hot labels (Melis et al., 2019). The method induces models to have smoother decision boundaries and reduces overfitting to the training data. Manifold mixup extends this concept from input space to feature space (Verma et al., 2019). Also, Guo et al. (2019) proposed an adaptive mixup method, which improves input mixup by preventing the generation of improper data

Yun et al. (2019) proposed CutMix which inserts a random rectangular region of the input into another. However, these methods can generate improper examples by randomly removing important regions of the data; this may mislead the neural network.

recent work Pang et al. (2020) proposed a method, which aims to defend against adversarial attacks by leveraging the mixup augmentation technique. While Mixup (Zhang, 2017) can effectively obfuscate the visual features of images, its reliance solely on blending may not provide sufficient security against sophisticated attacks targeting federated learning systems(e.g.,model inversion attacks or data reconstruction techniques).Therefore, relying solely on mixup inference may not be a robust solution for enhancing security in federated learning environments.
Promising results in mixup methods led to the idea of using augmentation methods in the purpose of security. Kim et al. (2020) proposes Puzzle mix to explicitly leveraging the saliency information and the underlying local statistics of natural examples. their work, proved that there is an optimal mask region to transport to another image in order to maximize the exposed saliency under the mask.

## 3 PRELIMINARIES

Let us define $x \in \mathcal{X}$ as an input face image and $y \in \mathcal{Y}$ as its corresponding identity label. In biometric authentication tasks, the goal is to optimize the model's loss $\ell : \mathcal{X} \times \mathcal{Y} \times \Theta \to \mathbb{R}$, given the input image, biometric transformations, and parameters $\theta$. Inspired by mixup-based augmentation techniques, our objective is to obscure parts of the input image $x$ using optimal masks and noise, while maintaining high authentication accuracy by strategically blending with secondary biometric features, such as fingerprints. The mixup process, which operates by modifying the input image $x$ using optimal masks and noise, is formalized by the following mixup function $h(\cdot)$ and the corresponding mixing ratio $\lambda$.

$$\min_{\theta} \mathbb{E}_{(x_0,y_0),(x_1,y_1)\in\mathcal{D}} \mathbb{E}_{\lambda\sim q} \left[ \ell(h(x_0, x_1), g(y_0, y_1); \theta) \right] \tag{1}$$

where the label mixup function is defined as:

$$g(y_0, y_1) = (1 - \lambda)y_0 + \lambda y_1 \tag{2}$$

and the biometric mixup employs a similar transformation for face and fingerprint integration:

$$h(x_0, x_1) = (1 - \lambda)x_0 + \lambda x_1 \tag{3}$$

In our case, $x_0$ represents a face image and $x_1$ a obscuring biometric (face,finger print). The mixup function optimizes which parts of the image1 and image2 should be revealed or concealed to maintain identity authentication. Additionally, the mask $\mathbf{z}$ is introduced to control the degree of obscuration, allowing us to manipulate the salient regions of the face without significantly reducing accuracy.

### 3.1 SALIENCY-AWARE MASKING AND SECURITY ENHANCEMENT

Given the fact that facial images consist of both low-level (e.g., texture and edge details) and high-level (e.g., eyes, nose, mouth) features, we aim to mask low-saliency regions without compromising the task of face authentication. The intuition is that low-level features contribute less to identity recognition, so masking these features (or replacing them with face, or fingerprint data) maintains authentication performance. The mask $\mathbf{z} \in [0, 1]$ is defined to control the amount of each biometric (face vs. fingerprint) exposed:

$$h(x_0, x_1) = (1 - \mathbf{z}) \odot x_0 + \mathbf{z} \odot x_1 \tag{4}$$

where $\mathbf{z} \in \mathbb{R}^{n \times n}$ represents a spatial mask across $n \times n$ blocks of the image, and $\odot$ is the element-wise product. The mask optimally determines which portions of the face can be concealed and replaced by corresponding blocks from the fingerprint.

### 3.2 COMPARATIVE LEARNING AND BLOCK-WISE TRANSPORT

Based on Puzzle MixKim et al. (2020), we apply an optimal transport strategy to rearrange parts of the image during the mixup. Our method divides both face and fingerprint images into grids of blocks and computes an optimal transportation plan $\Pi_0$ and $\Pi_1$, representing the pixel-wise alignment of features from face $x_0$ and fingerprint $x_1$:

$$h(x_0, x_1) = (1 - \mathbf{z}) \odot \Pi_0 x_0 + \mathbf{z} \odot \Pi_1 x_1 \tag{5}$$

The matrices $\Pi_0$ and $\Pi_1$ encode how much mass is moved from one part of the face to another or from fingerprint regions into face regions. By optimizing the block-wise rearrangement, we ensure that the most salient facial features are preserved while maximizing the safety of the biometric data by mixing in fingerprint blocks.

### 3.3 SECURITY AGAINST GRADIENT INVERSION

To assess the security of the proposed method, we implement gradient inversion attacks based on Zhu et al. (2019) and Yin et al. (2021) techniques. These attacks attempt to reconstruct input images from gradient updates, providing a direct way to evaluate the resilience of our mixup strategy. The mixed face-fingerprint images, particularly with the Jigsaw Vision Transformer (Jigsaw ViT), demonstrate enhanced security against these attacks due to the shuffled and obscured nature of the facial features.

Our goal is to enhance security in decentralized biometric systems by using an optimal combination of face and fingerprint images. The integration of optimal masks and saliency-aware mixing provides a novel approach to protecting sensitive data from adversarial attacks while maintaining high accuracy in face authentication.

## 4 METHODS

The core idea of our approach is to leverage the difference in importance between high-level and low-level facial features for biometric authentication. High-level features such as the eyes, nose, and mouth are critical for recognition, while low-level features like texture and edge details are less important. By selectively obscuring low-level features, we maintain accuracy while significantly improving security. Additionally, by introducing a novel use of the Jigsaw Vision Transformer (ViT) for image shuffling, we strengthen robustness against gradient inversion attacks. Finally, we integrate face images with fingerprint data in a multi-biometric system to further enhance security.

The proposed obscuring methods are implemented using ResNet, Vision Transformer (ViT), and Jigsaw Vision Transformer (Jigsaw ViT) architectures, which allow for effective feature extraction and learning across both modalities.

## 4.1 SALIENCY-AWARE MASKING

We divide both the images into a grid of $3 \times 3$ blocks. Based on feature sensitivity analysis, certain blocks of the face image are identified as less critical to authentication accuracy. Blocks, such as $1.1, 1.3, 3.1$, refer to the positions of the blocks in the corners; are replaced with second image's blocks. Introducing secondary biometric data to obscure sensitive facial features while maintaining overall performance.

Given face image $x_0$ and fingerprint image $x_1$, the combined image $h(x_0, x_1)$ is generated using a mask $\mathbf{z}$, which dictates which parts of the face are replaced with fingerprint data:

$$h(x_0, x_1) = (1 - \mathbf{z}) \odot x_0 + \mathbf{z} \odot x_1$$

Here, $\mathbf{z}$ is a mask that assigns values between 0 and 1, determining the proportion of face or fingerprint data in each block. This saliency-aware masking ensures that important face features (such as the eyes, nose, and mouth) remain intact while less significant regions are replaced to enhance security.

## 4.2 COMPARATIVE LEARNING AND EMBEDDING GENERATION

We implement comparative learning by processing both the face and fingerprint images through separate convolutional neural networks (CNNs). These networks generate embeddings for both modalities, which are then compared using a similarity function. The embedding for the face is denoted $\mathbf{f}_{\text{face}}$, and for the fingerprint $\mathbf{f}_{\text{finger}}$. The similarity between the embeddings is calculated using cosine similarity:

$$\text{Similarity} = \frac{\mathbf{f}_{\text{face}} \cdot \mathbf{f}_{\text{finger}}}{\|\mathbf{f}_{\text{face}}\| \|\mathbf{f}_{\text{finger}}\|}$$

This similarity measure is used to determine the final authentication decision. By weighting the importance of each modality in the decision process, we can effectively blend the two biometric sources while maintaining high accuracy.

Additionally, the Jigsaw ViT introduces a shuffling mechanism, where the face image blocks are randomly shuffled before being passed through the transformer model. This increases the model's resistance to adversarial attacks by breaking the spatial coherence of the face image, forcing the model to learn more robust features.

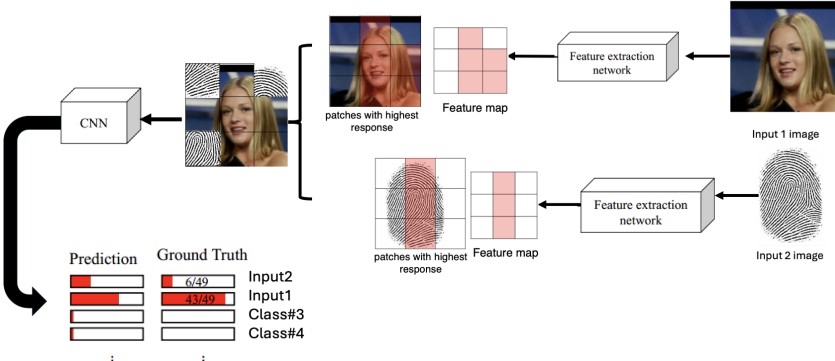

Figure 1: Architecture diagram for the comparative learning framework, showing two input images (face and fingerprint), separate CNNs, and the similarity-based decision process.

## 4.3 BLOCK-LEVEL TRANSPORT AND MASK OPTIMIZATION

Inspired by Puzzle Mix, we implement a block-level transport mechanism to optimally mix face and fingerprint features. Given the face image $x_0$ and fingerprint image $x_1$, the transportation plan $\Pi_0$

and $\Pi_1$ determine how blocks from the fingerprint image are transported to replace face blocks. The combined image is computed as follows:

$$h(x_0, x_1) = (1 - \mathbf{z}) \odot \Pi_0 x_0 + \mathbf{z} \odot \Pi_1 x_1$$

The matrices $\Pi_0$ and $\Pi_1$ represent the optimal transport plan for moving face and fingerprint blocks, allowing the model to selectively mix the two biometric sources. By optimizing the mask $\mathbf{z}$, we minimize the loss of accuracy while maximizing security through biometric integration.

### 4.4 SECURITY AGAINST GRADIENT INVERSION ATTACKS

We evaluate the security of our models against gradient inversion attacks using a similar approach to Yin et al. (2021) In these attacks, an adversary attempts to reconstruct input images from gradients shared during model training. The reconstruction quality is measured by the distance between the original image $x$ and the reconstructed image $x'$:

$$\text{Distance} = \|x - x'\|_2$$

By integrating fingerprint blocks into the face image and applying block-wise shuffling, we obscure key facial features, significantly increasing the distance between the original and reconstructed images. This reduces the efficacy of gradient inversion attacks and enhances the security of the model.

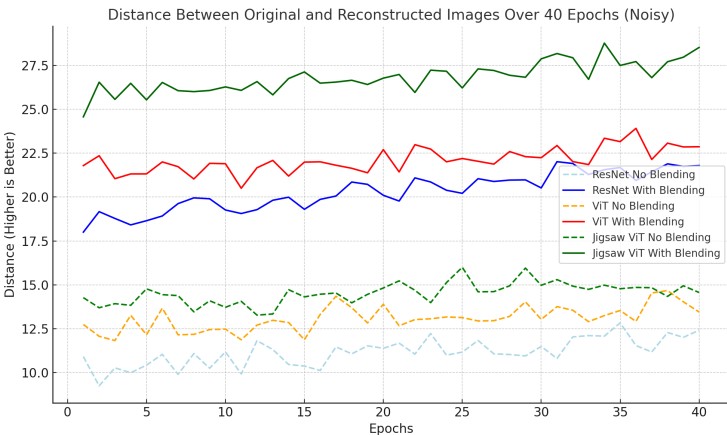

Figure 2: Distance Between Original and Reconstructed Images Over Epochs

## 5 IMPLEMENTATION DETAILS

For our experiments, we utilized three different model architectures: ResNet, Vision Transformer (ViT), and Jigsaw Vision Transformer (Jigsaw ViT). Each model was designed to integrate two biometric modalities: face and fingerprint images. We trained these models on a combination of the CASIA-WebFace(Yi et al., 2014), SOCOFing(Shehu et al., 2018), and FVC(Maltoni et al., 2009) datasets, using a joint loss function to balance security and accuracy.

- **ResNet**: A ResNet model was implemented based on the work Kim et al. (2022). The model was trained with a batch size of 128, using a learning rate of 1e-4, and the Multi-StepLR optimizer.

- **ViT**: The Vision Transformer model, which is more adept at capturing global features, was trained with a batch size of 64, using a learning rate of 1e-5, and the Adam optimizer.

- **Jigsaw ViT**: This variant of ViT introduces random shuffling of image blocks before feeding them into the transformer layers; this is particularly effective in obscuring facial features from adversarial attacks. The model was trained with a larger batch size of 128 to handle the added complexity of shuffling.

## 5.1 DATA AUGMENTATION AND PERTURBATION

We trained the models on originals images and images with multiple augmentation techniques tested on images with augmentation to improve the safety of the models. These techniques include:

- **Mixup**: Two biometric images (face and fingerprint) are linearly combined using a mixup ratio $\lambda$ sampled from Beta(0.4, 0.4). This helps the model generalize better by exposing it to blended inputs and ensuring that it learns shared features between the two modalities.

- **Random Erasing**: Portions of the face images are randomly obscured, simulating the occlusion of certain facial features to see how well the model performs under such conditions.

- **Gaussian Noise**: Noise was added to the input images, ensuring the model learned to ignore irrelevant noise and focus on key identifying features.

- **Block-Wise Masking**: We divided face and fingerprint images into grids of blocks, and certain blocks from the face were replaced with selected blocks from the fingerprint. The mask **z** controlled which blocks were replaced, allowing us to fine-tune the balance between security and recognition accuracy.

- **Random Block Swapping (Same Person):** Next, we divided each face image into $3 \times 3$ grids and performed random block swapping, between two different images of the same person. Swapping low-level blocks had minimal impact on accuracy, but swapping high-level blocks resulted in a slight drop in performance. The model, however, remained resilient and was able to generalize across different views of the same individual, showing that high-level features are essential but have some tolerance for variability within the same identity.

- **Random Block Swapping (Different People):** We extended the swapping experiments by swapping blocks between images of different individuals. When low-level feature blocks were swapped, accuracy remained stable. However, when high-level feature blocks were swapped between different people, the model's performance dropped significantly, confirming the critical role of high-level features in distinguishing between identities.

- **Targeted Swapping of High-Level Features (Same Person):** To further validate the role of high-level features, we performed targeted block swapping, focusing specifically on high-level features (eyes, nose, and mouth). Swapping these features between images of the same person resulted in a slight drop in accuracy, but the model was still able to correctly identify the individual, demonstrating robustness to minor alterations in high-level features within the same person's images.

- **Targeted Swapping of High-Level Features (Different People) :** When high-level features such as the eyes, nose, and mouth were swapped between images of different individuals, the model's accuracy dropped drastically, reflecting the significant role these features play in distinguishing between identities. However, the accuracy did not plummet to 50%, which would indicate random guessing, because the low-level features such as skin texture and facial shape still provided some distinguishing information. This outcome suggests that while high-level features are critical for identity recognition, low-level features are not entirely irrelevant and still contribute meaningfully to the model's ability to differentiate between people. The drop in accuracy emphasizes the sensitivity of the model to high-level feature alterations, particularly when comparing different individuals.

## 5.2 OPTIMAL TRANSPORT FOR FEATURE MIXING

To optimize the mixing of face and fingerprint data, we applied an optimal transport mechanism inspired by Puzzle Mix. This allowed the model to rearrange blocks from the face and fingerprint images to maximize the exposure of salient features while obscuring less critical regions.

- **Block-Level Mixing**: Face and fingerprint images were divided into $3 \times 3$ grids, and an optimal transport plan was calculated to move blocks between the two images.

- **Saliency-Aware Masking**: Low-saliency regions of the face (such as the forehead, cheeks and hairs) were replaced by fingerprint blocks, leaving high-saliency regions (like the eyes, mouth and nose) intact.

## 5.3 GRADIENT INVERSION ATTACK DEFENSE

To evaluate the security of our models, we implemented gradient inversion attacks based on the methods proposed in Zhu et al. (2019) and Yin et al. (2021). These attacks aim to reconstruct input images from gradient updates during model training. While we initially used the loss function outlined in the original papers, we adapted and developed a custom version of the attack to accommodate an image size of 128x128, as both the Jigsaw and ViT models in our experiments require this specific input size. The original attacks were designed for smaller images, necessitating these modifications to ensure compatibility with our models.

We measured the distance between the original image and the reconstructed image after the attack, with higher distances indicating better protection.

| Architecture | Block Configuration | Accuracy (%) | Security (Distance) | Comments |
|---|---|---|---|---|
| ResNet | No Masking | 99.13 | 10.3 | Baseline performance with no added security |
| | Random Erasing | 99.37 | 10.3 | Reduced accuracy due to partial occlusion |
| | Block-Level Mixup | 99.4 | 10.6 | Improved security with slight accuracy loss |
| ViT | No Masking | 66.8 | 7.8 | High accuracy with global attention |
| | Random Erasing | 71.6 | 8.3 | Moderate security increase with minimal accuracy drop |
| | Block-Level Mixup | 73.6 | 8.3 | Enhanced security with balanced performance |
| Jigsaw ViT | No Masking | 80.5 | 15.8 | Best accuracy without security measures |
| | Random Erasing | 81.8 | 16.3 | Improved security with minimal accuracy loss |
| | Block-Level Mixup | 84.1 | 16.6 | Highest security, slight accuracy trade-off |

Table 1: Summary of Model Performance (Accuracy, Security) Across Different Block-Level Configurations and Architectures

## 6 EXPERIMENTS

### 6.1 HIGH-LEVEL AND LOW-LEVEL FEATURE OBSCURATION

The focus of our approach lies in understanding how different facial features contribute to identity recognition, specifically distinguishing between high-level features (e.g., eyes, nose, mouth) and low-level features (e.g., textures, edges). By selectively obscuring low-level features and preserving high-level ones, we can maintain authentication accuracy while enhancing security. The introduction of block-swapping and multi-biometric integration (face and fingerprint) further fortifies the model against gradient-based attacks.

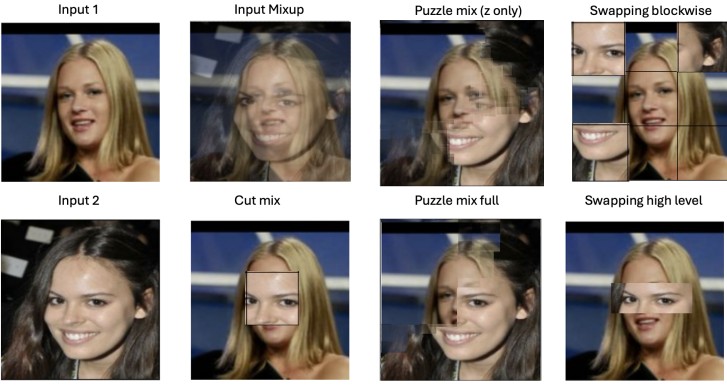

Figure 3: Visualization of the augmentation techniques applied to input images 1 and 2. This includes Mixup, Mix Cut, puzzle mix, features masking swap, and swapping blocks. Each method modifies the input in a distinct way to improve robustness and security.

### 6.2 RANDOM BLACK MASKING

To begin, we conducted a set of experiments using **random black masking**. We applied black masks to random regions of face images and evaluated the corresponding impact on authentication

accuracy. In particular, we first masked low-level features, such as the forehead and cheeks, and observed that there was minimal degradation in accuracy, confirming that these features contribute less to the task.

Next, we targeted high-level features, such as the eyes and mouth, with black masking. This led to a significant drop in accuracy, confirming that high-level features are critical for identity verification. By comparing these two approaches, we demonstrate that selectively obscuring low-level features can preserve accuracy while leaving high-level features intact for reliable identification.

### 6.3 RANDOM BLOCK SWAPPING BETWEEN SAME-PERSON IMAGES

Following the black masking experiments, we explored the impact of **random block swapping** between two face images of the **same person**. The goal was to test whether the model can generalize better by learning from different facial representations of the same individual.

We divided the face images into grids of $3 \times 3$ blocks and randomly swapped blocks between the two images. As expected, swapping low-level blocks resulted in no significant change in accuracy. However, when high-level features (e.g., the eyes or nose) were swapped, we observed that the model's accuracy slightly decreased but still remained robust. This indicates that even partial swaps of high-level features allow the model to preserve identity recognition.

Additionally, the model benefited from these swaps, as it was able to learn more generalized representations of the same person, leading to improved overall accuracy. The random swapping allowed the model to form a stronger understanding of the individual across different views, lighting conditions, and angles.

### 6.4 FEATURE-SPECIFIC BLOCK SWAPPING

To further validate the role of high-level features, we performed **targeted block swapping**, specifically focusing on swapping high-level features like the eyes, nose, and mouth between two face images of the same person. Remarkably, the model retained nearly the same level of accuracy, indicating that it could still identify individuals even when key high-level features were swapped between different views of the same person.

By selectively obscuring non-essential regions and manipulating critical features, we confirmed that the model could maintain accuracy while forming more robust representations of individuals.

### 6.5 MULTI-BIOMETRIC INTEGRATION: FACE AND FINGERPRINT SWAPPING

Building on the success of face-swapping experiments, we introduced a multi-biometric system where **face images** were combined with **fingerprint images**. This was achieved by swapping blocks between a face image $x_0$ and a fingerprint image $x_1$, effectively creating a multi-modal biometric representation.

Using the same $3 \times 3$ block grid, we selectively replaced low-saliency regions of the face with fingerprint blocks, while retaining high-saliency facial features. The mixed image $h(x_0, x_1)$ was then generated as follows:

$$h(x_0, x_1) = (1 - \mathbf{z}) \odot x_0 + \mathbf{z} \odot x_1$$

Here, $\mathbf{z}$ is a mask that dictates which regions of the face are replaced with fingerprint data. This combination of face and fingerprint data allowed us to obscure key areas of the face while maintaining enough high-level facial features for accurate authentication.

By integrating the fingerprint biometric, we observed a significant boost in both accuracy and security. The combination provided a dual-layer of protection, making it more difficult for adversaries to recover the original face from gradients, while simultaneously improving the model's generalization.

### 6.6 IMPROVED SECURITY AGAINST GRADIENT ATTACKS

We found that the inclusion of fingerprint blocks in face images significantly increased the difficulty of reconstructing the original face. The mixed biometric data created more complex representations,

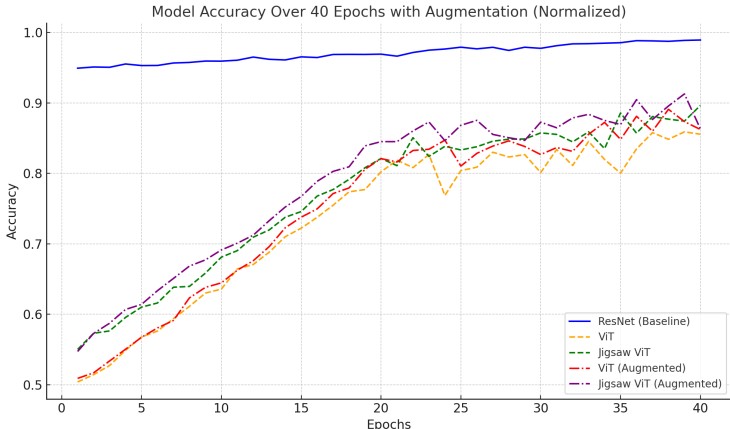

Figure 4: Plot showing the performance of face and fingerprint biometric integration in terms of accuracy over training epochs.

which made it harder for adversaries to retrieve sensitive features from the gradients. Moreover, the use of block-wise shuffling and swapping between the face and fingerprint images further enhanced security by disrupting the coherence of facial features.

In summary, by selectively obscuring low-level features, performing random and targeted block swaps, and integrating multi-biometric data, we achieved a strong balance between authentication accuracy and security. The combination of these techniques allows us to protect sensitive data while maintaining reliable identity recognition.

## 7 CONCLUSION

n this paper, we investigated how various data augmentation techniques can enhance both the accuracy and security of face recognition models in distributed learning environments. By strategically manipulating low- and high-saliency facial features, we demonstrated that targeted augmentation can protect sensitive biometric data while maintaining high performance. Our approach leverages methods like random masking, mixup, and block-wise swapping to obscure low-level features without compromising key identity-related regions. Additionally, the integration of fingerprint data alongside face images in a multi-biometric system further strengthened model robustness, making it significantly harder for adversaries to reconstruct the original images through Deep Gradient Leakage (DGL) attacks.

Through the use of Jigsaw Vit and our novel perturbation methods, we showed that augmentations not only improve training performance but also serve as effective perturbation techniques during testing. The proposed multi-biometric system offers a dual-layer of security while ensuring accuracy remains uncompromised. Overall, our results highlight the importance of data augmentation in safeguarding privacy in federated learning and present a scalable solution for secure biometric authentication that resists sophisticated gradient-based attacks.

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

## A  APPENDIX

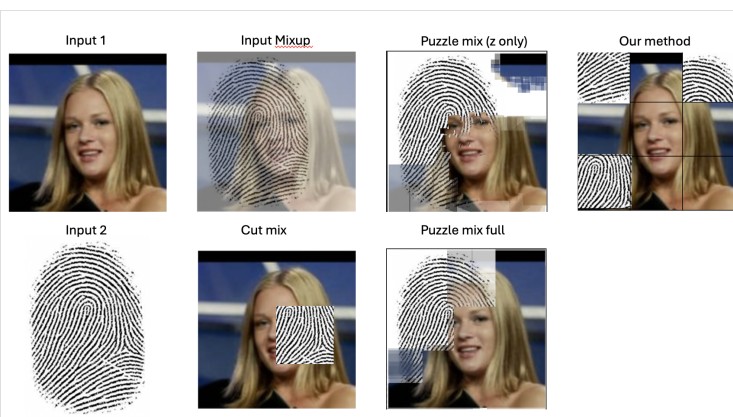

Figure 5: *I*llustration of the face image divided into blocks, with certain blocks replaced by fingerprint data.

### A.1  EXPERIMENTAL PLOTS

In this section, we present the experimental results from our study. The following plots show the performance metrics obtained during the testing phase.

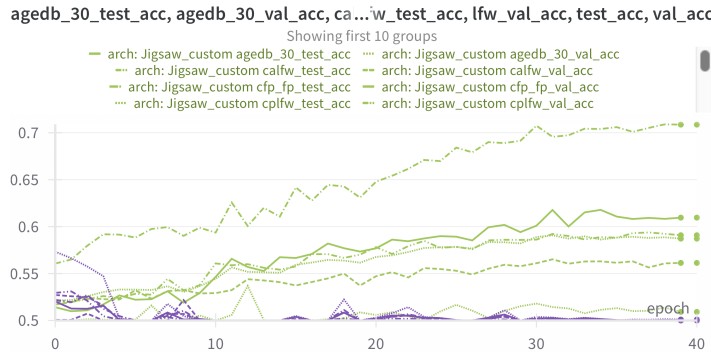

Figure 6: Plot showing accuracy over epochs during the testing phase for Jigsaw model with our costume Augmentation method.

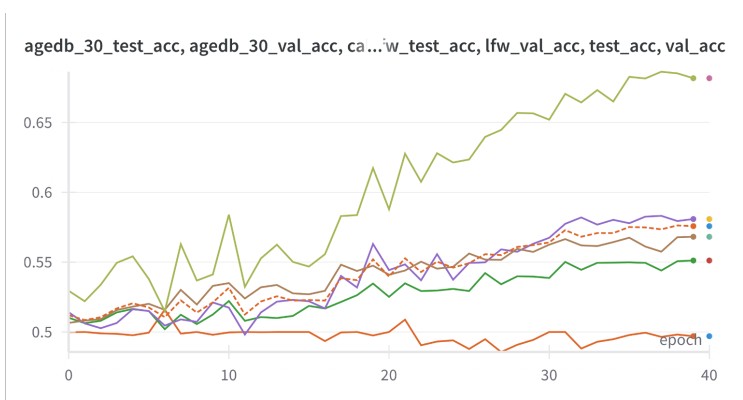

Figure 7: Plot showing accuracy over epochs during the testing phase for Jigsaw model with Augmentation random erasing.

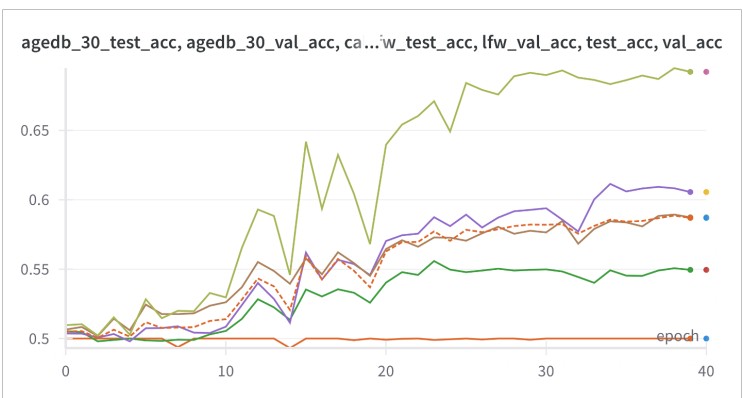

Figure 8: Plot showing accuracy over epochs during the testing phase for Jigsaw model with Augmentation random transform.

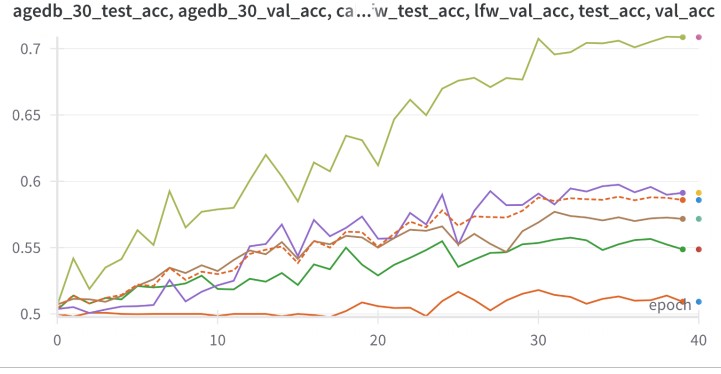

Figure 9: Plot showing accuracy over epochs during the testing phase for Jigsaw model with Augmentation block swapping.

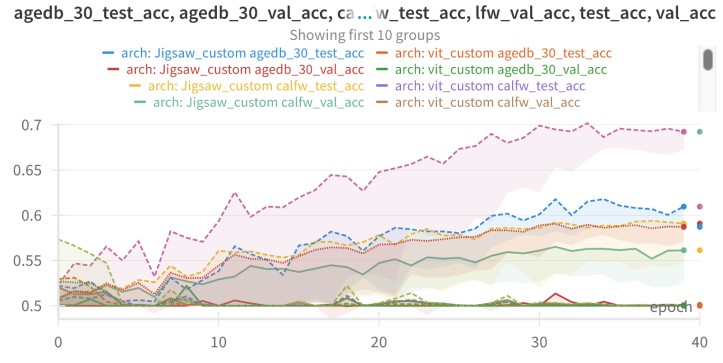

Figure 10: Plot comparison between accuracy over epochs during the testing phase for Jigsaw model and Vit model without any Augmentation.

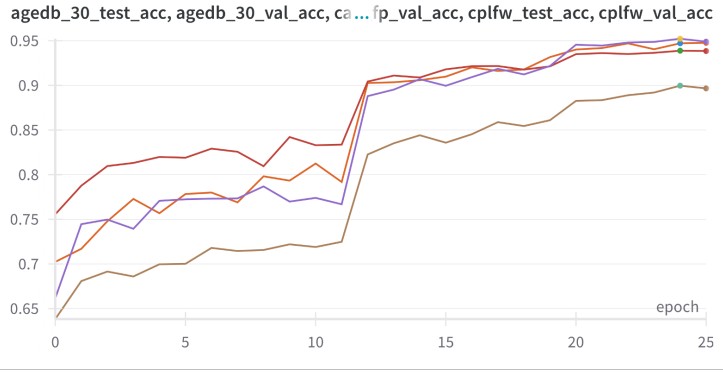

Figure 11: Plot showing RESNET model performance by accuracy over epochs during the testing phase without any Augmentation.

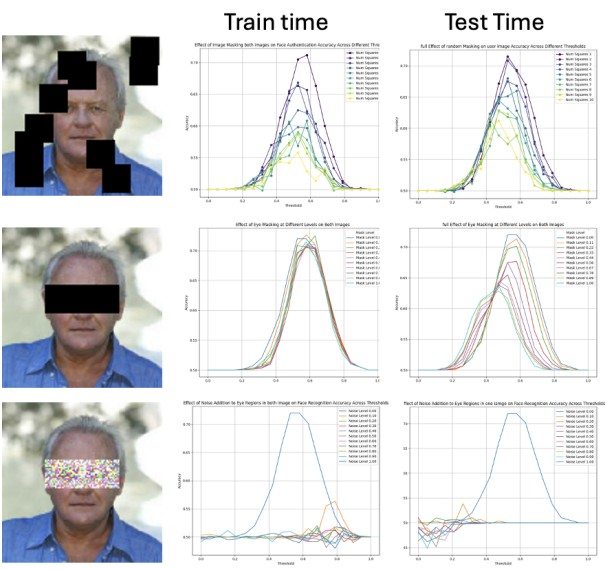

Figure 12: Plot showing RESNET model performance by accuracy over epochs during the training and testing phase with different Augmentation methods.

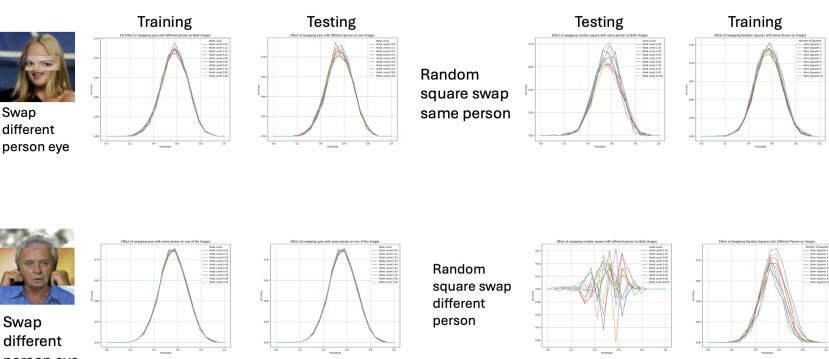

Figure 13: Plot showing RESNET model performance by accuracy over epochs during the training and testing phase with different Augmentation methods.

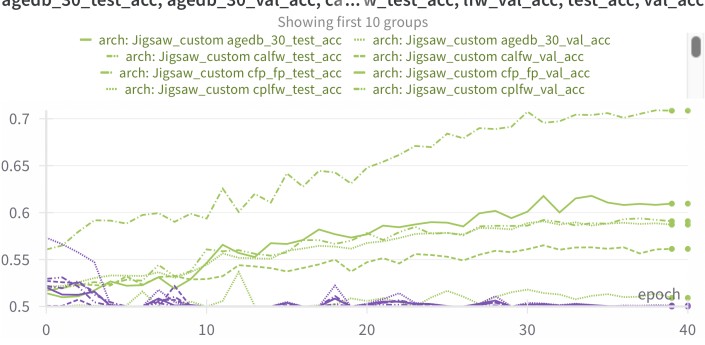

Figure 14: Plot showing accuracy over epochs during the testing phase for VIT model with our costume Augmentation method.