# OpenReview forum: "Gradients protection in federated learning for Biometric authentication"
_ICLR.cc/2025/Conference — ICLR 2025 Conference Withdrawn Submission_

### Official Review · Reviewer_VZuV · 2024-10-16

**Soundness:** 1
**Presentation:** 1
**Contribution:** 2
**Rating:** 1
**Confidence:** 4

**Summary:**

The paper presents a learning algorithm aimed at defending against model inversion attacks by protecting gradients and incorporating data augmentation techniques.

**Strengths:**

The approach of fusing face images and fingerprints for defending against model inversion attacks appears novel.

**Weaknesses:**

Major concerns:

1. The paper requires substantial improvements in writing, as the current version hinders readability.

2. The motivation for protecting gradients is debatable, as recent model inversion techniques do not necessarily rely on gradient information.
Zhang, H., Dong, X., Lai, Y., Zhou, Y., Zhang, X., Lv, X., ... & Li, X. (2024). Validating Privacy-Preserving Face Recognition under a Minimum Assumption. In Proceedings of the IEEE/CVF Conference on Computer Vision and Pattern Recognition (pp. 12205-12214).

3. The proposed methodology does not introduce new techniques; rather, it appears to be a combination of existing learning algorithms, architectures, and data augmentation strategies.

4. It is challenging to understand how the various components of the proposed method are connected, as well as their intended purpose and the effects they produce.

5. The methodology and implementation rely heavily on established methods, but lack any theoretical analysis or mathematical justification for selecting these approaches.

6. Neither the security nor the clean accuracy shows any notable improvements.

7. It is unclear how the authors conducted comparisons with benchmarks. There are no comparisons against defenses, nor were any ablation studies performed.

8. The attack methods used are outdated (see comment 2).

Suggestions on Writing and Evaluation Design:

a. The motivation needs to be clearly defined. The paper should introduce the model inversion attack scenarios (evaluation scenarios), outline their potential consequences, and describe the proposed defense strategies (contributions), explaining why they are effective. Additionally, the paper should include assumptions about the knowledge of both the attacks and the defenses (threat models).

b. While the proposed methodologies can be derived from existing works, each component must be accompanied by theoretical analysis (ideally with mathematical proofs) and demonstrated through experiments or ablation studies. This will help explain the rationale behind selecting these components/algorithms. It would also be beneficial to provide reasoning for the choices of implementation, including the datasets, benchmarks, and experimental settings.

c. The evaluation should be built on a solid baseline and compared against the latest benchmarks, for example, 2024 model inversion attacks and defenses. As this is a defense paper, the proposed strategy should demonstrate its generalizability across various types of attacks, rather than being limited to a specific one like DGL.

d. Research should not only present results, but also analyze and discuss them, drawing meaningful insights—especially when the outcomes do not align with theoretical expectations.

e. A summary of key information lacking or confusing in the current version: The paper should clarify the attack and defense scenarios, threat models, novelty, theoretical analysis, evaluation metrics, performance outcomes, ablation studies, and limitations.

**Questions:**

See Weaknesses.

---

### Official Review · Reviewer_vzgg · 2024-10-18

**Soundness:** 2
**Presentation:** 1
**Contribution:** 2
**Rating:** 1
**Confidence:** 5

**Summary:**

The paper addresses privacy challenges in face recognition models by using data augmentation to protect training images from Deep Gradient Leakage (DGL) attacks. It aims to enhance security and accuracy in distributed learning systems through four objectives: developing a privacy-focused model, using augmentation for safety, protecting against attacks, and introducing a novel perturbation method for biometric authentication.

**Strengths:**

The article focuses on the security of Deep Learning, especially Federated Learning, which is very valuable for current academia and industry.

**Weaknesses:**

1. There are huge problems with the writing, format, and references of this paper, which makes it very inconvenient and uncomfortable to read. For example, the first letter of the sentence is not capitalized (lines 29, 55, 134, 141, etc.), letters are missing (line 515), the reference format is incorrect and so on.

2. The method proposed in this paper lacks innovation and lacks comparison with existing benchmarks. InstaHide [1] is a classical defense that mixes multiple images and uses pixel-level encryption. In the current version, I haven't see where yours method are more advanced, and there are not enough experiments to prove it.

3. Although the authors claim that the proposed method can protect gradients, the paper lacks comprehensive experiments, both qualitative and quantitative, against Gradient Inversion Attacks (GIAs) like DLG [2], IG [3], etc.

[1] Huang, Yangsibo, et al. "Instahide: Instance-hiding schemes for private distributed learning." International conference on machine learning. PMLR, 2020.

[2] Zhu, Ligeng, Zhijian Liu, and Song Han. "Deep leakage from gradients." Advances in neural information processing systems 32 (2019).

[3] Geiping, Jonas, et al. "Inverting gradients-how easy is it to break privacy in federated learning?." Advances in neural information processing systems 33 (2020): 16937-16947.

**Questions:**

1. In your strategy (as shown in Figure 3), most of the eyes and facial details are still clearly visible. How does this work against Gradient Inversion Attacks (GIAs)?

2. The InstaHide method also uses data augmentation, but Carlini et al. [4] have pointed out the problems in instance-encoding. How does your method solve this limitation?

3. What loss function did you use during the training phase? In addition, why did you use backbone CNN models such as ResNet and ViT instead of face recognition models such as FaceNet and ArcFace?

4. In GIAs, people typically use PSNR, LPIPS or MSE to evaluate the performance of image reconstruction. I suggest you use these metrics directly instead of defining "Security (Distance)".

5. There are many formatting issues in the paper, such as the first letter of the sentence is not capitalized (lines 29, 55, 134, 141, etc.), letters are missing (line 515), the reference format is incorrect, etc. These problems do not meet the writing standards of a high-quality paper.

[4] Carlini, Nicholas, et al. "Is private learning possible with instance encoding?." 2021 IEEE Symposium on Security and Privacy (SP). IEEE, 2021.

---

### Official Review · Reviewer_MC74 · 2024-11-01

**Soundness:** 3
**Presentation:** 3
**Contribution:** 2
**Rating:** 5
**Confidence:** 3

**Summary:**

This paper addresses privacy challenges in distributed learning systems, particularly targeting the threat posed by Deep Gradient Leakage (DGL). It explores data augmentation methods to enhance security by protecting original training images from gradient-based attacks while maintaining or improving model accuracy. The study introduces a novel perturbation method for multi-biometric authentication systems, utilizing Vision Transformers (ViTs) and proposing a mix of augmentation strategies to safeguard biometric data effectively.

**Strengths:**

1. The paper proposes a unique integration of data augmentation and perturbation methods to protect sensitive training data in a federated learning context, which is both novel and pertinent given the increasing concerns about privacy in machine learning.
2. It provides comprehensive experiments using various datasets and models (ResNet, ViT, Jigsaw ViT) to demonstrate the efficacy of the proposed methods in protecting against gradient inversion attacks, which substantiates the claims with empirical evidence.

**Weaknesses:**

1. The paper primarily focuses on empirical results and lacks a deep theoretical analysis of why certain augmentation methods work effectively against DGL attacks. Including such an analysis could strengthen the theoretical foundations of the proposed methods.
2. While the paper shows improvements on specific datasets and models, it does not extensively discuss the generalizability of the proposed methods across different federated learning setups or against different types of attacks.
3. Although the paper discusses enhancing security, it lacks a detailed analysis of how resilient the proposed methods are against more advanced or different types of attacks not covered in the experiments.
4. The study could benefit from a more detailed comparison with state-of-the-art methods, specifically discussing where it outperforms or falls short, to provide clearer insights into its relative advantages and limitations.

**Questions:**

See weaknesses above.

---

### Official Review · Reviewer_boKb · 2024-11-02

**Soundness:** 2
**Presentation:** 2
**Contribution:** 2
**Rating:** 3
**Confidence:** 5

**Summary:**

This paper employs augmentation methods to perturb private images and enhance the security of neural networks. A perturbation method is introduced for multi-biometric authentication systems, providing accurate recognition while ensuring security and anonymity of entities.

**Strengths:**

The paper addresses an important issue in federated learning - gradient privacy leakage, which is a current research hotspot with significant application value. It explores the role of data augmentation techniques in improving the model's generalization ability and security.

**Weaknesses:**

1-Lack of innovation. The proposed method is a combination of several existing methods, and the innovations are insufficient to support acceptance at the conference.

2-Low accuracy. The accuracy of the proposed method decreases significantly compared to scenarios without privacy protection, leading to poor usability.

3-Multi-biometric authentication systems may require users to provide more biometric data, which could affect user acceptance and raise privacy concerns.

**Questions:**

Please see Weaknesses.

---

### Note · Authors · 2024-11-20

I have read and agree with the venue's withdrawal policy on behalf of myself and my co-authors.